# RETHINKING FLAT MINIMA: SEEKING $\epsilon$-MAXIMA TOWARD BETTER GENERALIZATION

## ABSTRACT

Modern deep neural networks are often over-parameterized, leading to significant overfitting issues: achieving a near-zero training loss while potentially generalizing poorly. In response, by employing Sharpness-Aware Minimization (SAM), seeking flat minima has been widely adopted as a common belief for achieving a better generalization, heuristically assuming that model parameters located in low-curvature regions of the training loss landscape will induce the same low loss values over the underlying data distribution. However, considering the inscrutable geometric structure of the real data distribution loss landscape, flat minima may not be the only optimal solution. We question whether an alternative geometric structure of the training loss landscape could offer better generalization over the underlying data distribution. To formalize this, we propose to seek an $\epsilon$-Maxima point that achieves a loss value at least $\epsilon$ greater than all points within a punctured perturbation domain of a given radius. We demonstrate that seeking such a point by leveraging our novel optimization framework, $\epsilon$-MS, surpasses both SAM and SAM-based methods on standard generalization benchmarks. Moreover, in stronger generalization scenarios—including long-tailed recognition and single-domain generalization, $\epsilon$-MS exhibits clear advantages. In particular, it achieves state-of-the-art performance on standard generalization benchmarks and long-tailed recognition tasks, highlighting its promising generalization performance across diverse training scenarios.

## 1 INTRODUCTION

Improving the generalization performance for deep neural networks (DNNs) is one of the main tasks in the field of modern learning theory. Due to the serious over-parameterization for the network architecture (Zhang et al., 2016), the loss landscape of DNNs is highly non-convex, resulting in numerous global optima, serious overfitting, and leading to poor generalization performances. Researches (Keskar et al., 2016; Dziugaite & Roy, 2017; Jiang et al., 2020; Neyshabur et al., 2017; Dinh et al., 2017) suggest that flat minima, covered by uniformly low loss values, always lead to a better generalization performance. Oppositely, sharp minima, often with abrupt loss changes, always leads to a poor generalization performance. Inspired by this phenomenon, recent work by (Foret et al., 2020) proposed a dual optimization method called Sharpness-Aware Minimization (SAM). By perturbing the parameters before performing the gradient descent step, SAM effectively enhances generalization performance by minimizing sharpness. Recent studies (Kwon et al., 2021; Luo et al., 2024; Du et al., 2021; Li et al., 2024; Wen et al., 2022; Chen et al., 2023) have contributed to the advancement of SAM theoretically and empirically, developing precious theoretical insights and algorithms.

Although flat minima have long been linked to improved generalization, however, in modern highly over-parameterized DNNs, there may exist an alternative ideal target. The over-parameterization will lead to two main results: the serious overfitting issue and complex loss landscapes. Thus, practically, such models are capable of driving the training loss arbitrarily low, regardless of whether the solution lies at a sharp peak, a flat basin, or even a local maximum. Based on this insight, we wonder if there exists an alternative ideal geometric structure that can have a better generalization performance. We therefore introduce the notion of an *ideal geometric structure*, $\epsilon$-Maxima: a center point whose training loss is at least $\epsilon$ higher ($\epsilon \in \mathbb{R}$) than the highest (worst-case) loss within a punctured neighborhood (inner radius $q$, outer radius $\rho$). Intuitively, as sketched in Figure 1, when $\epsilon > 0$, an $\epsilon$-Maxima places the center on a small peak while many nearby perturbations lie on an

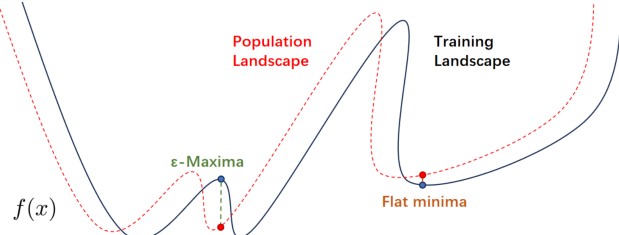

Figure 1: A Conceptual Sketch of $\epsilon$-Maxima and Flat Minima when $\epsilon \geqslant 0$. The Y-axis indicates the value of the loss function, and the X-axis indicates the parameters. The blue line denotes the training landscape, and the red dotted line is the population loss (true loss under the data distribution).

equal or lower moat. Under a train–population shift, such a configuration is attractive: A $\epsilon$-Maxima produced by constraining a $\epsilon$ margin may lead to a lower loss region on the population distribution. In short, we trade a bit of central training loss for a surrounding region that is uniformly better behaved, yielding a solution that is more likely to generalize.

Building on this idea, we develop $\epsilon$-Maxima Seeking ($\epsilon$-MS), which operationalizes the peak-with-moat geometry without complicating training. The $\epsilon$-MS pairs a SAM-style suppression of neighborhood worst-case loss with a lightweight and adaptive control that tempers the reduction rate of loss at the center point, realized through a simple proxy objective that is stable in practice and plug-and-play with standard optimizers. This proxy favors solutions that encourage a positive $\epsilon$ margin, making $\epsilon$-MS less prone to over-minimizing the training loss at the center. Theoretically, under standard smoothness assumptions we show the update admits a small, second-order remainder, provably enlarges the loss margin $\epsilon$, and benefits for a lower PAC-Bayesian generalization upper bound indexed by the punctured neighborhood; empirically, $\epsilon$-MS matches the computation of SAM while delivering stronger and more stable generalization across standard benchmarks, long-tailed recognition, and single-domain generalization.

To demonstrate the effectiveness of our $\epsilon$-MS algorithm and the idea of seeking an $\epsilon$-Maxima point, we conducted extensive experiments and compared our algorithm with SAM and SAM-based algorithms, which aim to find flat minima in different training scenarios. First, we train models from scratch and compare the generalization performance between our $\epsilon$-MS with SAM and SAM-based algorithms on CIFAR-10, CIFAR-100 (Krizhevsky et al., 2009) and ImageNet (Deng et al., 2009). Meanwhile, we compare $\epsilon$-MS with SAM under strong generalization tasks, including long-tailed learning and single-domain generalization. The experimental results align with our theoretical analysis and highlight that our method not only outperforms SAM but also provides more stable generalization across diverse conditions, providing a novel idealized target in over-parameterized regimes for better generalization performance.

## 2 Seeking $\epsilon$-Maxima for Better Generalization

**Notations**: We denote scalar as $a$, vector as $\boldsymbol{a}$ . From a distribution $\mathcal{D}$, we draw an i.i.d training dataset $\mathcal{S} \triangleq \bigcup_{i=1}^{n} \{(\mathbf{x}_i, \mathbf{y}_i)\}$, we seek to learn a model with high generalization ability. Consider a family of models parameterized by $\mathbf{w} \in \mathcal{W} \subseteq \mathbb{R}^d$; For a per-data-loss function $l : \mathcal{W} \times \mathcal{X} \times \mathcal{Y} \to \mathbb{R}_+$ we have the loss for training set: $L_S(\mathbf{w}) \triangleq \frac{1}{n} \sum_{i=1}^{n} l(\mathbf{w}, \mathbf{x}_i, \mathbf{y}_i)$ and the loss for the population: $L_{\mathcal{D}}(\mathbf{w}) \triangleq \mathbb{E}_{(x,y)\sim\mathcal{D}} [l(\mathbf{w}, \mathbf{x}, \mathbf{y})]$ . We only observed the training set $\mathcal{S}$, the goal of model training is to select the optimal model parameters $\mathbf{w}$ to have the lowest population loss $L_{\mathcal{D}}$.

### 2.1 Revisiting Sharpness-aware Minimization

The idea of Sharpness-Aware Minimization is to seek the model parameter $\mathbf{w}$ with uniformly low training loss value. The optimization problem of SAM can be described as follows:

$$\min_{\mathbf{w}} L_S^{\text{SAM}}(\mathbf{w}) + \lambda \|\mathbf{w}\|_2^2 \quad \text{where} \quad L_S^{\text{SAM}}(\mathbf{w}) \triangleq \max_{\|\boldsymbol{\delta}\|_p \leqslant \rho} L_S(\mathbf{w} + \boldsymbol{\delta}), \tag{1}$$

Here, $\rho > 0$ is a hyperparameter with $p \in [1, \infty]$, $\lambda \|\mathbf{w}\|_2^2$ yields a standard L2 regularization term and $\lambda$ is a hyperparameter. For the inner maximization problem $\max_{\|\boldsymbol{\delta}\|_p \leqslant \rho} L_S(\mathbf{w} + \boldsymbol{\delta})$, SAM use

the first-order Taylor expansion of $L_S(\mathbf{w} + \boldsymbol{\delta})$ to approximate the optimal value of $\boldsymbol{\delta}$ that is used for finding the neighbor that induces the highest loss value and represented as :

$$\hat{\boldsymbol{\delta}}(\mathbf{w}) = \rho \, \text{sign}\big(\nabla_{\mathbf{w}} L_S(\mathbf{w})\big) \frac{\big|\nabla_{\mathbf{w}} L_S(\mathbf{w})\big|^{v-1}}{\|\nabla_{\mathbf{w}} L_S(\mathbf{w})\|_v^{v/p}}, \tag{2}$$

where $1/p + 1/v = 1$, $|\cdot|$ denotes the element-wise absolute value and power, the adoption of the L2 norm ($v = p = 2$) in typical implementations leads to the degenerated form: $\hat{\boldsymbol{\delta}}(\mathbf{w}) = \rho \, \frac{\nabla_{\mathbf{w}} L_S(\mathbf{w})}{\|\nabla_{\mathbf{w}} L_S(\mathbf{w})\|_2}$. In practice, as shown in Eq.2, SAM uses the maximum loss on the sphere to approximate the worst-case loss in the neighborhood of $\mathbf{w}$. The actual optimization problem for SAM can be described as:

$$\min_{\mathbf{w}} \Phi_S(\mathbf{w}) + \lambda \|\mathbf{w}\|_2^2. \tag{3}$$

Here, $\Phi_S(\mathbf{w})$ denotes the worst-case loss on the sphere with radius $\rho$ where:

$$\Phi_S(\mathbf{w}) := \max_{\|\delta\|=\rho} L(\mathbf{w} + \delta). \tag{4}$$

Then, substituting the $\hat{\delta}$ calculated from equation 2 back into equation 1 and dropping the second-order terms, the final gradient can be approximated by:

$$\nabla_{\mathbf{w}} L_S^{\text{SAM}}(\mathbf{w}) \approx \nabla_{\mathbf{w}} \Phi_S(\mathbf{w}). \tag{5}$$

More detailed theoretical details can be found in the original study (Foret et al., 2020).

## 2.2 Seeking $\epsilon$-Maxima for Better Generalization Performance

DNNs are often over-parameterized, which often leads to an overfitting problem. In general, most of the time, regardless of the sharpness of the surrounding landscape, the model can always have a low loss performance on the training set. Compare to a sharp minima point, a worst point in a region with similar low loss performance on the training set may have better generalization performance due to the maxima. To formalize this, in this section, we will introduce our $\epsilon$-***Maxima*** and corresponding $\epsilon$-***Maxima Seeking ($\epsilon$-MS)*** algorithm.

### 2.2.1 $\epsilon$-Maxima

We first describe the proposed ideal geometric structure pursued during the optimization process as the $\epsilon$-Maxima structure. Inspired by the $\epsilon$ family in optimization and variational analysis (Mavrotas, 2009; Ehrgott & Ruzika, 2008), we present a general form of $\epsilon$-Maxima: *a center point $w$ is an $\epsilon$-**Maxima** when its loss value is at least $\epsilon$ higher than the highest loss proposed by others in its punctured neighborhood*, as shown in Figure 1. Leveraging the insight of approximating the worst-case loss in SAM, a parameter $w$ is an $\epsilon$-Maxima if :

$$L(w) - \max_{\delta: \, q \leqslant \|\delta\| \leqslant \rho} L(w + \delta) \geqslant \epsilon, \qquad \epsilon \in \mathbb{R}, \tag{6}$$

where $q \leqslant \|\delta\| \leqslant \rho$ defines a punctured neighborhood of $w$ by puncturing a center region (e.g., eliminating the area bounded by a small constant $q$) in the original neighborhood in SAM. When $\epsilon \geqslant 0$, the center behaves like a local maxima point in neighborhood, when $\epsilon < 0$, we capture a tolerant near-maximum.

In practical training scenarios, due to the limited number of training samples, there is an inevitable distribution discrepancy between the training set $\mathcal{S}$ and the population set $\mathcal{D}$. As indicated in SAM (Foret et al., 2020), reducing the worst-case loss in the neighborhood of a parameter $w$ can effectively lower the PAC-Bayesian generalization upper bound for population set $\mathcal{D}$ and result in better generalization performance. Similarly, under the standard Lipschitz-smoothness condition, by introducing above $\max_{q \leqslant \|\delta\| \leqslant \rho} L_S(w + \delta)$ term, we can rewrite the generalization upper bound as follows:

$$L_D(w) \leqslant L_S(w) - \left[ L_S(w) - \max_{q \leqslant \|\delta\| \leqslant \rho} L_S(w + \delta) \right] + \left( G \, q + \frac{\beta}{2} \, q^2 \right) + \mathcal{C}_{\text{PB}}, \tag{7}$$

where $\left( G \, q + \frac{\beta}{2} \, q^2 \right)$ is the smoothness term and $\mathcal{C}_{\text{PB}}$ denotes the PAC-Bayes complexity term. Detailed derivations are shown in Appendix A.6.

In Eq.7, the practical optimizable part is the first two term. Since the central loss $L_S(w)$ term can always be successfully reduced for modern over-parametrized models. Thus, a more important part of minimizing such an upper bound is to *keep a relatively large margin term* $[L_S(w) - \max_{q \leqslant \|\delta\| \leqslant \rho} L_S(w + \delta)]$. Such an observation aligns well with our motivation, seeking $\epsilon$-Maxima in Eq.6. In this paper, we propose a simple and effective solution to seek $\epsilon$-Maxima by directly keeping the reduction rate of the central loss $L_S(w)$ slower than other worst-case loss $\max_{q \leqslant \|\delta\| \leqslant \rho} L_S(w + \delta)$ in the punctured neighborhood.

### 2.2.2 OPTIMIZING OBJECTIVE

Although our ideal geometric structure is to have a large margin $\epsilon$ and preferably a positive one, however, for optimizability and numerical stability, especially to prevent infeasible constraints and divergence early on or under noise, we adhere to the classical $\epsilon$-constraint framework in the optimization field: *$\epsilon$ is formulated to admit both positive and negative values, thereby ensuring feasibility and enhancing robustness during training.*

To achieve the worst-case loss in the neighborhood of a specific parameter $\mathbf{w}$, SAM (Foret et al., 2020) approximates the highest loss on the sphere surface, i.e., $\Phi_S(\mathbf{w})$ in Eq.4 where $\|\delta\| = \rho$. Considering that our domain (e.g., a punctured neighborhood) is a subset of SAM, our worst-case loss is always smaller than SAM, $\max_{q \leqslant \|\boldsymbol{\delta}\|_p \leqslant \rho} L_S(\mathbf{w} + \boldsymbol{\delta}) \leqslant \max_{\|\boldsymbol{\delta}\|_p \leqslant \rho} L_S(\mathbf{w} + \boldsymbol{\delta})$. Thus, it is rational to approximate the worst-case loss in a punctured neighborhood by follow the Eq.3 in SAM (Foret et al., 2020). Correspondingly, our optimization objective of seeking $\epsilon$-Maxima is defined as follows:

$$\min_{\mathbf{w}} \Phi_S(\mathbf{w}) + \lambda \|\mathbf{w}\|_2^2 \quad s.t. \quad L_S(\mathbf{w}) - \Phi_S(\mathbf{w}) \geqslant \epsilon. \tag{8}$$

The above constrained optimization enforces a dual objective: *it minimizes the worst-case loss $\Phi_S(\mathbf{w})$ in a punctured neighborhood, enhancing robustness, while simultaneously ensuring the center point $\mathbf{w}$ satisfies an $\epsilon$-Maxima condition through the constraint.*

Applying a Lagrangian relaxation to the inequality constraint, we obtain the following unconstrained objective:

$$\hat{\mathcal{J}}(\mathbf{w}) = \Phi_S(\mathbf{w}) + \lambda \|\mathbf{w}\|_2 - k \left( L_S(\mathbf{w}) - \Phi_S(\mathbf{w}) - \epsilon \right), \tag{9}$$

where $k \geqslant 0$ is the Lagrange multiplier associated with the $\epsilon$-Maxima constraint. As can be observed, the essence of this Lagrange-type objective function is actually to impose a constraint to explicitly control the relative descent rate of the central loss $L_S(\mathbf{w})$ versus its worst-case loss $\Phi_S(\mathbf{w})$. Motivated by this, we do not explicitly track the multiplier in Eq.9, we use a stable surrogate optimization objective $\mathcal{J}$ that directly modulates the relative descent rate via an adaptive coefficient $\alpha$ (detailed in the following section). This yields the following proxy objective:

$$\mathcal{J}(\mathbf{w}) = \Phi_S(\mathbf{w}) - \alpha L_S(\mathbf{w}). \tag{10}$$

With such a proxy optimization objective and the adaptive $\alpha$, we demonstrate that the margin term for improving generation ability in Eq.7 can be effectively increased. More details are shown in the Appendix. In summary, we show that employing our conceptual goal, seeking $\epsilon$-Maxima in Eq.6, is able to lead to a lower generalization upper bound shown in Eq.7 to improve the generalization ability. The constrained formulation of Eq.8 admits a Lagrangian relaxation (Eq.9) leading to an optimizable proxy objective $\mathcal{J}(\mathbf{w})$. The final gradient used for updating can be expressed as:

$$\nabla_w \mathcal{J}(\mathbf{w}) \approx \underbrace{\nabla_w L_S(w + \hat{\delta}(\mathbf{w}))}_{g_n} - \alpha \underbrace{\nabla_w L_S(w)}_{g_c}, \tag{11}$$

where $\hat{\delta}$ is obtained by approximating the inner maximization in Eq.2 that drops the second-order term to further accelerate the computation. It can be observed that *such an update explicitly takes an action to reduce the neighborhood loss while ensuring the central loss decreases at a relatively slower rate to achieve an $\epsilon$-Maxima.* In fact, the $\epsilon$-MS update can be summarized by a leading first-order term, while higher-order contributions are neglected. Precise discussion—showing that, under a Lipschitz-smooth condition, the remainder admits an explicit $O(\eta^2)$ bound appears in Appendix A.4.

### 2.2.3 CONSTRAINING $\alpha$ TO MAINTAIN RELATIVELY SLOWER DESCENT OF THE CENTRAL LOSS

As mentioned before, rather than setting a fixed margin $\epsilon$ to seek $\epsilon$-Maxima, we use adaptive $\alpha$ to ensure that the central loss $L_S(\mathbf{w})$ can reduce relatively slower than the worst-case loss $\Phi_S(\mathbf{w})$ in the punctured neighborhood. Each update step is shown as follow:

$$\Delta \mathbf{W} = -\eta(g_n - \alpha g_c), \tag{12}$$

where $\eta$ represents the step length for each update. Let $\Delta L_n$ and $\Delta L_c$ denote the reduction of $\Phi_S(\mathbf{w})$ and $L_S(\mathbf{w})$ respectively within one update step, and can be denoted as:

$$\Delta L_c \approx \langle \Delta \mathbf{W}, g_c \rangle = -\eta(\langle g_n, g_c \rangle - \alpha \|g_c\|_2^2), \quad \Delta L_n \approx \langle \Delta \mathbf{W}, g_n \rangle = -\eta(-\alpha \langle g_n, g_c \rangle + \|g_n\|_2^2). \tag{13}$$

Then, the difference between the two reduction values is represented by $\Delta\Delta$ :

$$\Delta\Delta = \Delta L_c - \Delta L_n = \eta[\|g_n\|_2^2 - (1 + \alpha)\langle g_n, g_c \rangle + \alpha \|g_c\|_2^2]. \tag{14}$$

Since the objective is to ensure that $\Delta L_c$ has a relatively slower reduction rate than $\Delta L_n$, the difference term $\Delta\Delta$ should have a nonnegative value. Denote $r = \frac{\|g_c\|}{\|g_n\|}, \cos\theta = \frac{\langle g_n, g_c \rangle}{\|g_n\|\|g_c\|}$, we can rewrite the equation $\Delta\Delta \geqslant 0$ as follow:

$$\|g_n\|_2^2 - (1 + \alpha)\langle g_n, g_c \rangle + \alpha \|g_c\|_2^2 \geqslant 0,$$
$$1 - (1 + \alpha)r \cos\theta + \alpha r^2 \geqslant 0. \tag{15}$$

Thus, the $\alpha$ should satisfy the following conditions:

$$\Delta\Delta > 0 \iff \begin{cases} \alpha > \dfrac{r \cos\theta - 1}{r(r - \cos\theta)}, & \text{if } r > \cos\theta; \\[3mm] \alpha < \dfrac{r \cos\theta - 1}{r(r - \cos\theta)}, & \text{if } r < \cos\theta. \end{cases} \tag{16}$$

Notice that when $r = \cos\theta$, the left side of the Eq.15 is equal to $1 - \cos^2\theta$ which is nonnegative, thus for all $\alpha$, $\Delta\Delta \geqslant 0$. By introducing a threshold $\alpha_{thr} = \frac{r \cos\theta - 1}{r(r - \cos\theta)}$ and a small margin $\alpha_{mar}$, we calculate the raw value of $\alpha$ as:

$$\alpha_{raw} = \begin{cases} \alpha_{thr} + \alpha_{mar}, & \text{if } r > \cos\theta; \\[2mm] \alpha_{thr} - \alpha_{mar}, & \text{if } r < \cos\theta; \\[2mm] 0 \end{cases} \tag{17}$$

For the stability of training, we choose the minimum value of $\alpha$ as 0 and set $\alpha_{max}$ as a hyperparameter to clip the $\alpha_{raw}$ to get $\alpha_{final}$ as follow:

$$\alpha_{final} = \min(\max(\alpha_{raw}, 0), \alpha_{max}). \tag{18}$$

Since our method, like SAM, already requires computing both the central gradient $g_c$ and the neighborhood gradient $g_n$ through two backward passes, the additional step of calculating $\alpha_{final}$ incurs virtually no extra computational overhead and does not increase the overall training complexity. The detailed algorithm can be found in Appendix 1.

### 2.2.4 ON THE EFFECT OF ADAPTIVE $\alpha$ FOR LOSS REDUCTION DYNAMICS

Conceptually, under standard Lipschitz-smoothness and a small-step condition, our proxy objective in Eq.10, together with the $\epsilon$-MS updates, can provably maintain a positive $\Delta\Delta$ term to increase the margin, effectively optimize the intended true objective; full details are shown in Appendix A.5. Empirically, to verify that our adaptive $\alpha$ implementation can lead to a positive $\Delta\Delta$ term so that an increase of the margin term can be obtained, we record the value of $\Delta\Delta$ for our $\epsilon$-MS in each epoch and compared it with SAM in Figure 2. Notably, $\epsilon$-MS maintains positive $\Delta\Delta$ values across nearly all epochs, while SAM exhibits fluctuating $\Delta\Delta$ values. This observation demonstrates the effectiveness of our proposed adaptive $\alpha$ for increasing the margin value practically, which potentially leads to a lower upper bound in Eq.7.

## 3 EXPERIMENTS

To demonstrate the effectiveness of our proposed $\epsilon$-MS algorithm, following (Foret et al., 2020; Kwon et al., 2021; Du et al., 2021; Luo et al., 2024; Li et al., 2024), we apply our algorithm on CIFAR-10, CIFAR-100, and ImageNet from scratch. Moreover, we do extra experiments on strong generalization scenarios, including single domain generalization and long-tail learning. In all cases, we measure the generalization ability of $\epsilon$-MS by simply replacing SAM. As the results shown in the following chapter, $\epsilon$-MS behaves obviously stronger generalization performance than SAM and other baseline methods in different generalization scenarios. **More details of our experiments can be found in the Appendix, e.g., settings of hyperparameters for each experiment A.10, additional experiments including robustness to perturbation radius A.7, fine-tuning on downstream tasks A.8, sensitivity analysis A.9.**

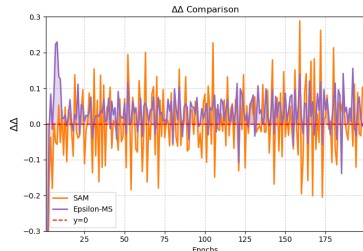

Figure 2: $\Delta\Delta$ term of each training epoch for SAM and $\epsilon$-MS on CIFAR-100 with ResNet-18

### 3.1 Image Classification From Scratch

In this section, we evaluate the generalization ability of $\epsilon$-MS on CIFAR-10, CIFAR-100 and ImageNet.

**CIFAR-10 and CIFAR-100** We start by applying our $\epsilon$-MS algorithm on CIFAR-10 and CIFAR-100 (Krizhevsky et al., 2009) image classification tasks. Following (Du et al., 2021; Luo et al., 2024; Li et al., 2024; Foret et al., 2020), the evaluation is carried out on three different architectures: ResNet-18 (He et al., 2016), WideResNet-28-10 (Zagoruyko & Komodakis, 2016), and PyramidNet-110 (Han et al., 2017). For fair comparison, we set all the training settings, including the training epochs, data augmentations, iterations and so on. Besides SAM, we additionally select SAM-based algorithms, including ESAM (Du et al., 2021), ASAM (Kwon et al., 2021), F-SAM (Li et al., 2024), as our extra baselines. As the results shown in Table 1, our $\epsilon$-MS algorithm significantly enhances model generalization performance on both CIFAR-10 and CIFAR-100 (Krizhevsky et al., 2009) datasets. Specifically, for CIFAR-10, $\epsilon$-MS outperforms SAM and even all the SAM-based baselines on all three backbones. For CIFAR-100, our $\epsilon$-MS demonstrated significant performance improvements: $\epsilon$-MS outperforms 1.71, 1.65 and 1.68 points compared to SAM and even outperforms 1.14, 0.56 and 0.58 points compared to SAM-based algorithms on three backbones. This demonstrates the effectiveness of seeking $\epsilon$-Maxima for better generalization performance.

**ImageNet** To evaluate the effectiveness of our $\epsilon$-MS algorithm on a large-scale dataset, we conduct experiments on ImageNet (Deng et al., 2009), which contains 1000 classes and more than 1.2 million training images. We apply our method on ImageNet with ResNet-50 and ResNet-101 as our backbones. For the sake of fairness, we train 90 epochs with the same training settings for SGD, SAM and our $\epsilon$-MS algorithm. As shown in Table 2, our $\epsilon$-MS algorithm outperforms SAM by 0.81 and 0.80 points and even outperforms ESAM by 0.46 and 0.31 points on two backbones. This demonstrates the effectiveness of our $\epsilon$-MS algorithm on large-scale dataset.

Table 1: Classification accuracies on the CIFAR-10 and CIFAR-100 datasets. We use the same code base as (Du et al., 2021) with the same data augmentations. For the sake of fairness, we reproduce the results of F-SAM (Li et al., 2024) with the same data augmentation.

| CIFAR-10 | SGD | ASAM | ESAM | F-SAM | SAM | $\epsilon$-MS (ours) |
|---|---|---|---|---|---|---|
| ResNet-18 | $94.51_{\pm0.03}$ | $96.57_{\pm0.15}$ | $96.56_{\pm0.08}$ | $96.58^{*}_{\pm0.06}$ | $96.52_{\pm0.13}$ | $\mathbf{96.62_{\pm0.11}}$ |
| WRN-28-10 | $96.34_{\pm0.12}$ | $97.33_{\pm0.13}$ | $97.29_{\pm0.11}$ | $97.35^{*}_{\pm0.14}$ | $97.27_{\pm0.11}$ | $\mathbf{97.54_{\pm0.06}}$ |
| PyramidNet-110 | $96.62_{\pm0.10}$ | $97.44_{\pm0.11}$ | $97.81_{\pm0.01}$ | $97.46^{*}_{\pm0.11}$ | $97.30_{\pm0.10}$ | $\mathbf{97.96_{\pm0.07}}$ |

| CIFAR-100 | SGD | ASAM | ESAM | F-SAM | SAM | $\epsilon$-MS (ours) |
|---|---|---|---|---|---|---|
| ResNet-18 | $78.17_{\pm0.05}$ | $80.74_{\pm0.12}$ | $80.41_{\pm0.13}$ | $80.71^{*}_{\pm0.22}$ | $80.17_{\pm0.17}$ | $\mathbf{81.88_{\pm0.13}}$ |
| WRN-28-10 | $81.56_{\pm0.13}$ | $83.60_{\pm0.24}$ | $84.51_{\pm0.01}$ | $83.88^{*}_{\pm0.14}$ | $83.42_{\pm0.04}$ | $\mathbf{85.07_{\pm0.12}}$ |
| PyramidNet-110 | $81.89_{\pm0.17}$ | $84.50_{\pm0.11}$ | $85.56_{\pm0.05}$ | $85.52^{*}_{\pm0.08}$ | $84.46_{\pm0.04}$ | $\mathbf{86.14_{\pm0.10}}$ |

Table 2: Classification accuracies on the ImageNet dataset.

|  | ResNet-50 | ResNet-101 |
|---|---|---|
| SGD | 76.00 | 77.80 |
| ESAM | 77.05 | 79.09 |
| SAM | 76.70 | 78.60 |
| $\epsilon$-MS | **77.51** | **79.40** |

Table 3: Results on CIFAR-10-LT and CIFAR-100-LT with different imbalance factors.

| Method | CIFAR-10-LT | | CIFAR-100-LT | |
|---|---|---|---|---|
| | 100 | 50 | 100 | 50 |
| BBN (Zhou et al., 2020) | 79.9 | 82.2 | 42.6 | 47.1 |
| KCL (Kang et al., 2020) | 77.6 | 81.7 | 42.8 | 46.3 |
| TSC (Li et al., 2022) | 79.7 | 82.9 | 43.8 | 47.4 |
| HCL (Wang et al., 2021) | 81.4 | 85.4 | 46.7 | 51.9 |
| RIDE (3 experts) (Wang et al., 2020) | 81.6 | 84.0 | 48.6 | 51.4 |
| ETF-DR (Yang et al., 2022) | 76.5 | 81.0 | 45.3 | 50.4 |
| RBL (Peifeng et al., 2023) | 84.7 | 87.6 | 53.1 | 57.2 |
| ARB (Xie et al., 2023) | 83.3 | 85.7 | 47.2 | 52.6 |
| CE* | 75.4 | 78.3 | 42.1 | 48.1 |
| CE* + SAM | $76.83_{+0.12}$ | $79.25_{+0.13}$ | $43.77_{+0.10}$ | $49.19_{+0.08}$ |
| CE* + $\epsilon$-MS | $77.92_{+0.12}$ | $79.77_{+0.11}$ | $45.02_{+0.14}$ | $50.02_{+0.11}$ |
| GLMC (Du et al., 2023) | 87.8 | 90.2 | 55.9 | 61.1 |
| GLMC + MaxNorm (two-stage) | 87.6 | 90.2 | 57.1 | 62.3 |
| GLMC + SAM | $92.23_{+0.30}$ | $92.37_{+0.43}$ | $58.23_{+0.37}$ | $63.48_{+0.36}$ |
| GLMC + $\epsilon$-MS | $\mathbf{92.44}_{+0.31}$ | $\mathbf{92.55}_{+0.36}$ | $\mathbf{59.02}_{+0.42}$ | $\mathbf{64.44}_{+0.32}$ |

## 3.2 STRONG GENERALIZATION SCENARIOS

To further evaluate the effectiveness of seeking a $\epsilon$-Maxima point, we apply our $\epsilon$-MS algorithm on two strong generalization scenarios: Single domain generalization(SDG) and long-tail learning.

### 3.2.1 LONG-TAIL LEARNING

Data in real-world scenarios always follow a long-tail distribution, where the head classes dominate the sample space while the tail classes only have a few samples. In long-tail, the training dataset are long-tail distributed with a balance factor $\beta = N_{max}/N_{min}$ where N denotes the number of samples for a specific class. Our target is to generalize the model performance from an imbalanced training dataset to a balanced testing dataset. Following standard long-tail training protocol, we choose ResNet-32 as our backbone and apply $\epsilon$-MS on CIFAR-10-LT and CIFAR-100-LT (Cui et al., 2019) dataset with two different imbalance factors [100,50]. We also compare our method with different strong baseline methods. As the result shown in Table 3 our $\epsilon$-MS consistently achieves the best results on all datasets and imbalanced factors. Specifically, we outperform the current state-of-the-art (SOTA) algorithm GLMC (Du et al., 2023) 3.12, 3.34 points on the CIFAR-100-LT dataset. Moreover, compared to SAM (Foret et al., 2020), we continuously outperform 0.79 and 0.96 points and achieve the SOTA performance on long-tail training tasks.

### 3.2.2 SINGLE DOMAIN GENERALIZATION

Single domain generalization aims to train a robust model on a single source domain to against unknown target domain shifts. In this section, we evaluate the generalization ability of our $\epsilon$-MS algorithm. We compare our $\epsilon$-MS with SAM on the PACS (Li et al., 2017) and the Office-Home (Venkateswara et al., 2017) dataset using ERM++ (Teterwak et al., 2025) as our baseline. For PACS dataset, we use the Photo domain as the source domain and evaluate model performance on the Art, Cartoon, and Sketch domains. For Office-Home dataset, we use Realworld domain as the source domain and evaluate model performance on the Art, Clipart, Product domains. For a fairness

Table 4: Single domain generalization results on PACS (left) and Office-Home (right).

| Method | A | C | S | Avg. | Clip-art | Art | Product | Avg. |
|--------|---|---|---|------|----------|-----|---------|------|
| ERM++ | 65.43 | 32.59 | 47.82 | 48.61 | 48.73 | 55.88 | 73.02 | 59.21 |
| SAM | 67.08 | 34.51 | 46.32 | 49.30 | 49.19 | 56.41 | 73.49 | 59.70 |
| $\epsilon$-MS | **68.16** | 33.20 | **51.67** | **51.03** | **49.51** | **57.46** | **74.60** | **60.40** |

comparison, we ensure all algorithms are applied within the same 50 training epochs with the same data augmentation strategies. We use ResNet-18 as the training backbone network. In Table 4, we report the experiment results of each target domains and the mean accuracy across all target domains. Notably, our $\epsilon$-MS outperforms SAM on both PACS and Office-Home datasets, demonstrating the effectiveness of seeking an $\epsilon$-Maxima in the parameter space for better generalization performance.

# 4 DISCUSSIONS

## 4.1 COMPARISON BETWEEN $\epsilon$-MS AND SAM

Taking $\epsilon$-maxima as the ideal objective means we prioritize suppressing the neighborhood worst point while ensuring the decrease in the center with a relatively lower pace (targeting a higher, ideally positive margin $\epsilon$). Comparing with two different ideal structures, SAM did not impose any constraints to control the reduction rate of the central loss; our method, with an adaptive $\alpha$ in Eq.18, $\epsilon$-MS can have a positive $\Delta\Delta$ term for nearly all epochs and lead to a lower upper bound in Eq.7. To empirically observe the above phenomenon, we compare the SGD, SAM, and $\epsilon$-MS by plotting the accuracy gap and the training accuracy. The results align with our theoretical analysis and the above discussions. Specifically, as shown in Figure 3(b), $\epsilon$-MS also achieves high training accuracy, which is consistent with our premise that modern over-parameterized DNNs typically drive the training loss to a low region. Based on this observation, as shown in Figure 3(a), $\epsilon$-MS has a significantly lower generalization gap, demonstrating the effectiveness of seeking the ideal geometry of $\epsilon$-Maxima.

## 4.2 TRAINING DYNAMICS OF $\epsilon$-MS: NO RESISTANCE TO THE DECLINE OF THE CENTER

In our $\epsilon$-MS optimization framework, although the constraints require the center towards an $\epsilon$-Maxima, this does not imply the center is ignored or resisted during the optimization process. In fact, the center gradient still plays a crucial role during training. When we expand the gradient $\nabla_{\mathbf{w}} \mathcal{J}(\mathbf{w})$ with respect to $\mathbf{w}$ while treating $\hat{\delta}$ as a constant (matching with our implementation):

$$\nabla_{\mathbf{w}} L(\mathbf{w} + \hat{\delta}) - \alpha \nabla_{\mathbf{w}} L_S(\mathbf{w}) \approx (1 - \alpha)\nabla_{\mathbf{w}} L_S(\mathbf{w}) + \mathcal{H}(\mathbf{w})\hat{\delta} + \mathrm{R}. \tag{19}$$

Here, $\mathrm{R}$ denotes the remainder and $\mathcal{H}(\mathbf{w}) = \nabla_{\mathbf{w}}^2 L_S(\mathbf{w})$ is the Hessian. We can observe that the gradient for the center is not eliminated but preserved with a weight of $(1 - \alpha)$. Moreover, this shrinkage in gradient weight prevents the excessively rapid minimization of training loss and mitigates the overfitting issue. From Figure 3(a) and (b), $\epsilon$-MS is more effective in mitigating overfitting, achieving stronger generalization performance. As less weight for central loss reduction, more attention for the optimization will transfer to the hessian term, which leads the center towards an $\epsilon$-Maxima point. To further investigate the geometry behavior of $\epsilon$-MS, we contrast the neighborhood loss heatmap with the same perturbation between SAM and $\epsilon$-MS. As shown in Figure 3(c) and (d), different from flat minima, $\epsilon$-MS is not the lowest loss point in the neighborhood; however, the worst-direction boundary losses are substantially reduced while the center loss is less aggressively minimized. This phenomenon aligns with our theoretical observation on the training dynamics of $\epsilon$-MS, leading to stronger robustness and generalization ability.

# 5 RELATED WORK

**Flat Minima** The discussion of the flat minima can be traced back to Hochreiter & Schmidhuber (1994). Due to the geometrical properties of a flat minima, the connection between the flat minima and generalization has been widely discussed Keskar et al. (2016); Dziugaite & Roy (2017); Jiang

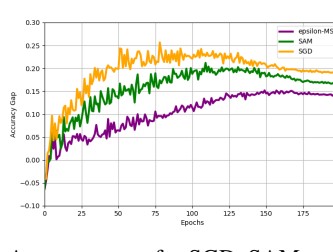 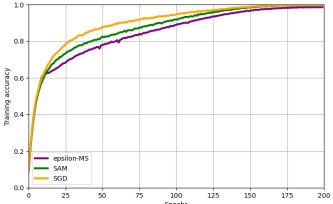

(a) Accuracy gaps for SGD, SAM and $\epsilon$-MS     (b) Training accuracy for SGD, SAM and $\epsilon$-MS

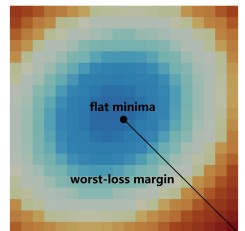 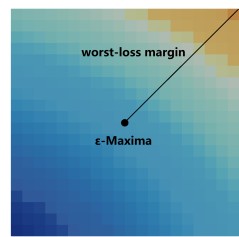

(c) Neighborhood loss heatmap for SAM     (d) Neighborhood loss heatmap for $\epsilon$-MS

Figure 3: Generalization behavior and the local loss geometry of $\epsilon$-MS and SAM (a) and (b) demonstrate the test-train accuracy gap and the training accuracy among SGD, SAM, and $\epsilon$-MS on CIFAR-100 trained with WideResNet-28-10. (c) and (d) are the neighborhood loss heatmaps with the same perturbation radius for SAM and $\epsilon$-MS that are trained by employing a ResNet. The centers of (c) and (d) are the origin; warmer colors indicate higher loss.

et al. (2020); Neyshabur et al. (2017); Dinh et al. (2017). Recently, many works have tried to improve the generalization performance through seeking flat minima (Zhang et al., 2024; Ahn et al., 2023; Zhao et al., 2022). For example, by minimizing local entropy, Chaudhari et al. (2019) proposed Entropy-SGD, enabling the model to converge to a flat region. Mobahi (2016) penalize the sharp minima, including operating on a diffuse loss landscape. Notably, sharpness-aware minimization (SAM) Foret et al. (2020), which leverages the connection between flat minima and generalization error, achieves significant success in achieving effective and efficient generalization. On this basis, numerous studies on SAM and related methods have emerged.

**Sharpness-Aware Minimization (SAM)**     Since the great success of SAM across various tasks, many studies have focus on construct a deeper understanding to SAM. For example, Wen et al. (2022) showed that SAM enhances the flatness of the minima by reducing the top eigenvalue of the Hessian in the full batch setting. Chen et al. (2023) attributed SAM's success on non-smooth convolutional ReLU networks to its capacity to suppress noise. In addition to theoretical explanations, many works have tried to improve the performance of SAM from different aspects. Kwon et al. (2021) improves SAM from the perspective of neighborhood geometry, Li et al. (2024) mitigates the negative effects of the full gradient components, Du et al. (2021; 2022) achieves a more efficient optimization process, and Zhuang et al. (2022) improves the SAM training through a surrogate loss.

## 6 CONCLUSION

In this paper, we rethink the conventional view of seeking flat minima for better generalization. We introduce the notion of $\epsilon$-Maxima, which might have stronger generalization performance under modern over-parameterization scenarios. Based on that, we proposed our $\epsilon$ Maxima Searching ($\epsilon$-MS) algorithm, which suppresses the worst-case behavior in the neighborhood while maintaining a controlled descent for the center loss. Both theoretical analysis and empirical evidence support the effectiveness of $\epsilon$-MS, suggesting that $\epsilon$-Maxima offer a new perspective for improving generalization in modern deep networks.

## 7 ETHICS STATEMENT

This research has been conducted in alignment with the ICLR Code of Ethics. We are committed to responsible stewardship of machine learning research, ensuring that our work advances knowledge while considering its potential societal impacts. In particular, we uphold high standards of scientific rigor, transparency, and reproducibility, and we affirm that no data has been falsified, fabricated, or misrepresented. Our study avoids harm by carefully considering possible negative consequences and by respecting privacy, fairness, and inclusiveness in the use of data and methods. All data used complies with relevant ethical approvals and license requirements, and precautions have been taken to prevent re-identification or misuse. We respect the intellectual contributions of others and provide appropriate credit where due. We believe this work contributes positively to human well-being by addressing problems of scientific and social relevance in ways that are transparent, responsible, and consistent with the principles of the ICLR Code of Ethics.

## 8 REPRODUCIBILITY STATEMENT

We have taken several steps to ensure the reproducibility of our work. The main experimental setup, including model architectures, training procedures, and evaluation metrics, is described in detail in the main paper and appendix. To facilitate reproducibility, we will release the majority of the code with an anonymous code link (shown in the Appendix) during the review process. If the paper is accepted, we commit to releasing the complete code base for all major experiments, along with detailed documentation and instructions for reproducing the reported results.

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

# A APPENDIX

## A.1 USAGE OF LARGE LANGUAGE MODELS

We use a large language model to polish the writing, correct grammar errors, and check typographical errors, thereby improving the overall accuracy and fluency of our paper.

## A.2 CODE

Our code can be found at https://anonymous.4open.science/r/epsilonMS-2A47/README.md

## A.3 ALGORITHM

---

**Algorithm 1** $\epsilon$-MS algorithm

---

**Input:** Training set $\mathcal{S} \triangleq \bigcup_{i=1}^{n}\{(\mathbf{x}_i, \mathbf{y}_i)\}$, Loss function $l : \mathcal{W} \times \mathcal{X} \times \mathcal{Y} \rightarrow \mathbb{R}_+$, Batch size $b$, Step size $\eta > 0$, Neighborhood size $\rho > 0$
**Output:** Model trained with $\epsilon$-MS
  1: Initialize weights $\mathbf{w}_0$, $t = 0$;
  2: **while** not converged **do**
  3:     Sample batch $\mathcal{B} = \{(\mathbf{x}_1, \mathbf{y}_1), \ldots, (\mathbf{x}_b, \mathbf{y}_b)\}$;
  4:     Compute gradient $g_c = \nabla_{\mathbf{w}} L_{\mathcal{B}}(\mathbf{w})$ of the batch's training loss;
  5:     Compute $\hat{\delta}(\mathbf{w})$ per Eq.2;
  6:     Compute gradient approximation for the $\epsilon$-MS objective
        $g_n = \nabla_{\mathbf{w}} L_{\mathcal{B}}(\mathbf{w})|_{\mathbf{w}+\hat{\boldsymbol{\delta}}(\mathbf{w})}$;
  7:     Compute $\alpha$ per Eq.18
  8:     Update weights: $\mathbf{w}_{t+1} = \mathbf{w}_t - \eta(g_n - \alpha g_c)$;
  9:     $t = t + 1$;
10: **end while**
11: **return** $\mathbf{w}_t$

---

## A.4 $\beta$-SMOOTH BOUND FOR THE REMAINDER TERM OF $\epsilon$-MS UPDATE

Assume the loss function is $\beta$-smooth, then for any point $x$ and step $u$, we have:

$$L(x + u) \leqslant L(x) + \langle \nabla L(x), u \rangle + \frac{\beta}{2}\|u\|^2, \quad L(x + u) \geqslant L(x) + \langle \nabla L(x), u \rangle - \frac{\beta}{2}\|u\|^2 \quad (20)$$

We apply the upper bound to $\Delta L_n$ and the lower bound to $\Delta L_c$ :

$$\Delta L_c \geqslant \langle g_c, \Delta \mathbf{W} \rangle - \frac{\beta}{2}\|\Delta \mathbf{W}\|^2, \quad \Delta L_n \leqslant \langle g_n, \Delta \mathbf{W} \rangle + \frac{\beta}{2}\|\Delta \mathbf{W}\|^2 \quad (21)$$

Thus, we can have:

$$\Delta L_c - \Delta L_n \geqslant \langle g_c - g_n, \Delta \mathbf{W} \rangle - \beta\|\Delta \mathbf{W}\|^2 \quad (22)$$

Symmetrically, when we take the lower bound to $\Delta L_n$ and the upper bound to $\Delta L_c$:

$$\Delta L_c \leqslant \langle g_c, \Delta \mathbf{W} \rangle + \frac{\beta}{2}\|\Delta \mathbf{W}\|^2, \quad \Delta L_n \geqslant \langle g_n, \Delta \mathbf{W} \rangle - \frac{\beta}{2}\|\Delta \mathbf{W}\|^2 \quad (23)$$

We can have:

$$\Delta L_c - \Delta L_n \leqslant \langle g_c - g_n, \Delta \mathbf{W} \rangle + \beta\|\Delta \mathbf{W}\|^2 \quad (24)$$

So we can get the upper bound of the remainder term:

$$\mathcal{R} = |(\Delta L_c - \Delta L_n) - \langle g_c - g_n, \Delta \mathbf{W} \rangle| \leqslant \beta\|\Delta \mathbf{W}\|^2 \quad (25)$$

Notice that $\Delta \mathbf{W} = -\eta(g_n - \alpha g_c)$, thus the remainder term can be rewritten as:

$$|\mathcal{R}| \leqslant \beta\eta^2\|g_n - \alpha g_c\|^2 = O(\eta^2) \quad (26)$$

## A.5 GROWTH OF THE LOSS MARGIN

Following the previous proof, we assume the loss function is $\beta$-smooth, and we have:

$$L_S(\mathbf{w} + \Delta\mathbf{W}) - L_S(\mathbf{w}) \geqslant \langle g_c, \Delta\mathbf{W}\rangle - \frac{\beta}{2}\|\Delta\mathbf{W}\|^2 \tag{27}$$

Then we upper bound the neighborhood worst term $\Phi$:

$$\Phi_S(\mathbf{w} + \Delta\mathbf{W}) - \Phi_S(\mathbf{w}) \leqslant \langle g_n, \Delta\mathbf{W}\rangle + \frac{\beta}{2}\|\Delta\mathbf{W}\|^2 \tag{28}$$

We then use $m(\mathbf{w})$ where $m(\mathbf{w}) = L_S(\mathbf{w}) - \Phi_S(\mathbf{w})$ to denote the loss margin between the center loss and the worst point in the neighborhood. Thus, for each update step, we have:

$$m(\mathbf{w} + \Delta\mathbf{W}) - m(\mathbf{w}) = (L_S(\mathbf{w} + \Delta\mathbf{W}) - L_S(\mathbf{w})) - (\Phi_S(\mathbf{w} + \Delta\mathbf{W}) - \Phi_S(\mathbf{w})) \tag{29}$$

and we can get the lower bound for this term:

$$m(\mathbf{w} + \Delta\mathbf{W}) - m(\mathbf{w}) \geqslant \langle g_c - g_n, \Delta\mathbf{W}\rangle - \beta\|\Delta\mathbf{W}\|^2 \tag{30}$$

Since $\Delta\mathbf{W} = -\eta(g_n - \alpha g_c)$, we can rewrite the lower bound in the previous equation and let it larger than 0:

$$\langle g_c - g_n, \Delta\mathbf{W}\rangle - \beta\|\Delta\mathbf{W}\|^2 = \eta(\|g_n\|^2 - (1 + \alpha)\langle g_n, g_c\rangle + \alpha\|g_c\|^2) - \beta\eta^2\|g_n - \alpha g_c\|^2 \tag{31}$$

Since we use apply the adaptive $\alpha$ to ensure the first term $\eta(\|g_n\|^2 - (1 + \alpha)\langle g_n, g_c\rangle + \alpha\|g_c\|^2)$ greater than 0. Thus, as long as:

$$\eta \leqslant \frac{(\|g_n\|^2 - (1 + \alpha)\langle g_n, g_c\rangle + \alpha\|g_c\|^2)}{\beta\|g_n - \alpha g_c\|^2}. \tag{32}$$

The margin m will increase after the update, demonstrating the validity of our proxy objectives and $\epsilon$-MS updates.

## A.6 PAC-BAYESIAN GENERALIZATION BOUND

In this section, we state a PAC-Bayesian Generalization Bound and demonstrate the effectiveness of our algorithm through this bound.

First, following (Foret et al., 2020), Let $S$ be a training set of size $n$, and $L_S(\cdot)$, $L_D(\cdot)$ be empirical and population risks. Fix a radius $\rho > 0$. For any parameter $w \in \mathbb{R}^k$, with probability at least $1 - \zeta$ over the draw of the training set $S \sim D$, the following holds:

$$L_D(w) \leqslant \max_{\|\delta\|_2 \leqslant \rho} L_S(w+\delta) + \frac{\sqrt{k\log\left(1 + \frac{\|w\|_2^2}{\rho^2}\left(1 + \sqrt{\frac{\log n}{k}}\right)^2\right) + 4\log\frac{n}{\zeta} + \underbrace{C\log(6n + 3k)}_{\tilde{O}(1)}}}{n - 1}. \tag{33}$$

Here $C > 0$ is an absolute constant (can be subsumed into $\tilde{O}(1)$), $n = |S|$, $k$ is the number of parameters. More detailed derivation and proofs of this upper bound are shown in (Foret et al., 2020).

Then, under the widely used $\beta$-smooth assumption, on the whole ball $\{z : \|z - w\| \leqslant \rho\}$, i.e., $\|\nabla L_S(z) - \nabla L_S(z')\| \leqslant \beta\|z - z'\|$. Assume also a gradient bound $G \geqslant \sup_{\|z-w\|\leqslant\rho}\|\nabla L_S(z)\|$. For any direction $u$ with $\|u\| = 1$ and $0 \leqslant r \leqslant q \leqslant \rho$, define $\varphi(t) := L_S(w + tu)$. Then

$$\varphi(r) \leqslant \varphi(q) + G(q - r) + \frac{\beta}{2}(q - r)^2. \tag{34}$$

Consequently,

$$\max_{\|\delta\|\leqslant q} L_S(w + \delta) \leqslant \max_{\|\delta\|=q} L_S(w + \delta) + G q + \frac{\beta}{2}q^2 \leqslant \max_{q\leqslant\|\delta\|\leqslant\rho} L_S(w + \delta) + G q + \frac{\beta}{2}q^2. \tag{35}$$

Thus, we have:

$$\max_{\|\delta\| \leqslant \rho} L_S(w + \delta) = \max\{\max_{\|\delta\| \leqslant q} L_S(w + \delta), \max_{q \leqslant \|\delta\| \leqslant \rho} L_S(w + \delta)\}$$

$$\leqslant \max\{\max_{q \leqslant \|\delta\| \leqslant \rho} L_S(w + \delta) + G\,q + \frac{\beta}{2}\,q^2, \max_{q \leqslant \|\delta\| \leqslant \rho} L_S(w + \delta)\}.\} \qquad (36)$$

So, we have:

$$\max_{\|\delta\| \leqslant \rho} L_S(w + \delta) \leqslant \max_{q \leqslant \|\delta\| \leqslant \rho} L_S(w + \delta) + G\,q + \frac{\beta}{2}\,q^2 \qquad (37)$$

We then can rewrite the previous upper bound as follows:

$$L_D(w) \leqslant \max_{q \leqslant \|\delta\| \leqslant \rho} L_S(w + \delta) + G\,q + \frac{\beta}{2}\,q^2 + \frac{\sqrt{k \log\left(1 + \frac{\|w\|_2^2}{\rho^2}\left(1 + \sqrt{\frac{\log n}{k}}\right)^2\right) + 4\log\frac{n}{\zeta} + \tilde{O}(1),}}{n - 1}$$

$$= L_S(w) - (L_S(w) - \max_{q \leqslant \|\delta\| \leqslant \rho} L_S(w + \delta)) + G\,q + \frac{\beta}{2}\,q^2 +$$

$$\frac{\sqrt{k \log\left(1 + \frac{\|w\|_2^2}{\rho^2}\left(1 + \sqrt{\frac{\log n}{k}}\right)^2\right) + 4\log\frac{n}{\zeta} + \tilde{O}(1),}}{n - 1} \qquad (38)$$

Our PAC-Bayes bound is indexed by the punctured neighborhood, and the trainable leading term in the upper bound is $L_S(w) - (L_S(w) - \max_{q \leqslant \|\delta\| \leqslant \rho} L_S(w + \delta))$ (since the smoothness correction depends only on q, $\rho$ and the smoothness constant, while the complexity term changes slowly) which matches the optimization objective we mentioned in the main paper. As we discuss in the previous section, our adaptive $\alpha$ and the proxy optimization objective can effectively increase the margin term $(L_S(w + \delta) - \max_{q \leqslant \|\delta\| \leqslant \rho} L_S(w + \delta))$ while the loss of the cental points is able to reduce to a relatively low value since the over-parameterized nature of the modern neural networks, our algorithm can effectively lower the upper bound in equation 38 theoretically and achieves better generalization performance(as shown in our experiment results) empirically. Also, as we set $q = 0$, the upper bound in equation 38 recovers the standard SAM case, underscoring the consistency.

### A.7 ROBUSTNESS TO PERTURBATION RADIUS

One main weakness of Sharpness-Aware Minimization (SAM) is its high sensitivity to the perturbation radius. Specifically, a relatively large perturbation radius may result in a significant decrease in generalization performance. In our $\epsilon$-MS, since we use an adaptive $\alpha$ to ensure the center loss decreases more slowly than the worst point in the punctured neighborhood, we expected our algorithm to be more robust to the choices of the perturbation radius. Thus, following (Foret et al., 2020; Li et al., 2024), we conduct experiment on CIFAR-100 with ResNet-18 to test the robustness of $\epsilon$-MS to perturbation radius. As shown in figure 3, $\epsilon$ is much less sensitive to $\rho$than SAM. When the perturbation is set 5 times larger than the optimal on CIFAR-100, the performance of SAM dropped from 80.17% to 78.80% while $\epsilon$-MS maintains a good performance of 81.77%. This confirms that compared to SAM, $\epsilon$-MS has a significant robustness improvement to perturbation radius.

### A.8 FINETUNING

Following (Luo et al., 2024; Foret et al., 2020; Li et al., 2024), we further evaluate the performance on fine-tuning tasks. In specific, we apply our algorithm on a ViT-B-16 model (Dosovitskiy et al., 2020) pretrained on ImageNet-1K for CIFAR-10 and CIFAR-100. We use the official checkpoint provided by the PyTorch repository. For fair comparison, we use the same initial learning rate of 0.01 for SGD, SAM and our $\epsilon$-MS algorithm and trained for 8k steps. We use the same radius for the perturbation term ($\rho = 0.05$) of SAM and our $\epsilon$-MS algorithm. For the hyperparameter of our algorithm, we set $\alpha_{max} = 0.6$ to maintain numerical stability. Table 5 shows the test accuracy where our $\epsilon$-MS algorithm consistently out performs the SGD and SAM, demonstrate the effectiveness of our algorithm on the fine-tuning task.

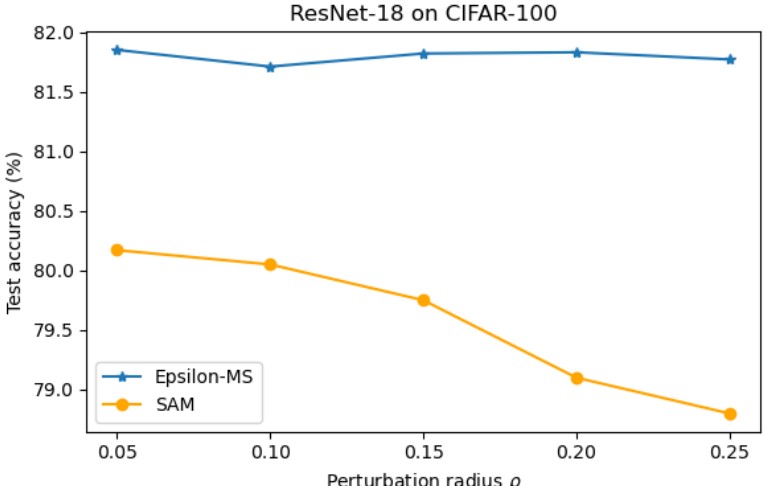

Figure 4: Enter Caption

Table 5: Test accuracy for fine-tuning ViT-B-16 pretrained on ImageNet-1K on CIFAR-10 and CIFAR-100.

| Architecture | Method | CIFAR-10 | CIFAR-100 |
|---|---|---|---|
| ViT-B-16 | SGD | $98.0 \pm 0.1$ | $88.4 \pm 0.1$ |
| | SAM | $98.3 \pm 0.1$ | $89.4 \pm 0.1$ |
| | $\epsilon$-MS | $\mathbf{98.6} \pm 0.1$ | $\mathbf{89.8} \pm 0.1$ |

## A.9 SENSITIVITY ANALYSIS

Since we use the hyperparameter $\alpha_{max}$ to clip the adaptive $\alpha$ for the stability of numerical calculation and training, here we conduct the sensitivity analysis of this hyperparameter. In detail, our experiments are conducted on the CIFAR-100 dataset and use ResNet-18 as our training backbone. We set the initial training rate as 0.05, weight decay as 0.001 and use a cosine learning rate scheduler.

As shown in Figure 5 for $\alpha_{max}$ we test the value of 0.75, 0.80, 0.85, 0.90, 0.95, 1.00. The results demonstrate the effectiveness and the stability of seeking $\epsilon$-Maxima point for better generalization. However, even though we can consistently achieve better performance than SAM, big $\alpha$ will lead to a performance decrease since large $\alpha$ may cause an under-fitting issue in the training due to the large suppression of the central loss decreasing rate. Meanwhile, too small $\alpha$ might cause a over-fitting issue and leads to a decrease in model weights.

## A.10 IMPLEMENTATION DETAIL

In this section, we will discuss the implementation details for our experiments. For all of our experiments, we apply our methods using the Py-Torch toolbox on GeForce RTX 4090 GPUs. All models are trained with by the SGD optimizer with a momentum of 0.9. Detailed hyperparameters are listed in Table 6 7 8 9.

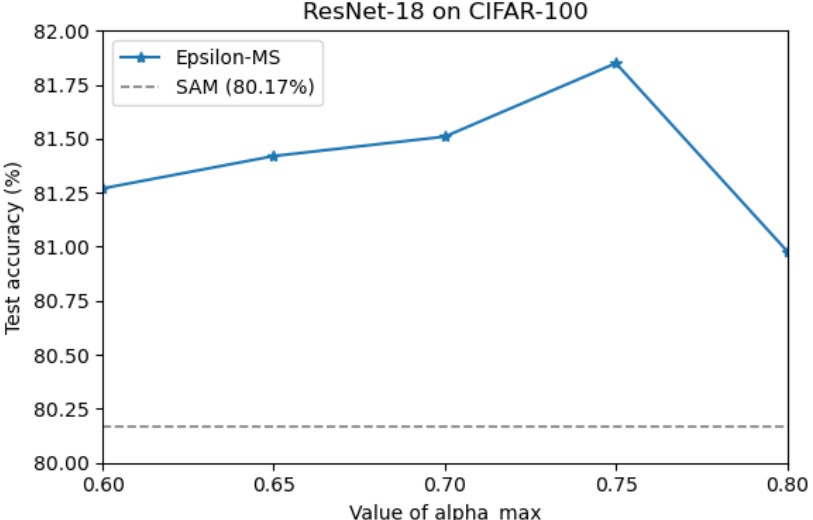

Figure 5: Sensitive analysis of $\alpha_{max}$

Table 6: Hyperparameters for training from scratch

| Model | | CIFAR-10 | | | CIFAR-100 | | |
|---|---|---|---|---|---|---|---|
| | | $\epsilon$-MS | SAM | ESAM | $\epsilon$-MS | SAM | ESAM |
| ResNet-18 | Epoch | 200 | 200 | 200 | 200 | 200 | 200 |
| | Batch size | 128 | 128 | 128 | 128 | 128 | 128 |
| | Data augmentation | Basic | Basic | Basic | Basic | Basic | Basic |
| | Peak learning rate | 0.05 | 0.05 | 0.05 | 0.05 | 0.05 | 0.05 |
| | Learning rate decay | Cosine | Cosine | Cosine | Cosine | Cosine | Cosine |
| | Weight decay | $1 \times 10^{-3}$ | $1 \times 10^{-3}$ | $1 \times 10^{-3}$ | $1 \times 10^{-3}$ | $1 \times 10^{-3}$ | $1 \times 10^{-3}$ |
| | $\rho$ | 0.05 | 0.05 | 0.05 | 0.05 | 0.05 | 0.05 |
| | $\alpha_{max}$ | 0.75 | 0.0 | 0.0 | 0.75 | 0.0 | 0.0 |
| Wide-28-10 | Epoch | 200 | 200 | 200 | 200 | 200 | 200 |
| | Batch size | 256 | 256 | 256 | 256 | 256 | 256 |
| | Data augmentation | Basic | Basic | Basic | Basic | Basic | Basic |
| | Peak learning rate | 0.05 | 0.05 | 0.05 | 0.05 | 0.05 | 0.05 |
| | Learning rate decay | Cosine | Cosine | Cosine | Cosine | Cosine | Cosine |
| | Weight decay | $1 \times 10^{-3}$ | $1 \times 10^{-3}$ | $1 \times 10^{-3}$ | $1 \times 10^{-3}$ | $1 \times 10^{-3}$ | $1 \times 10^{-3}$ |
| | $\rho$ | 0.1 | 0.1 | 0.1 | 0.1 | 0.1 | 0.1 |
| | $\alpha_{max}$ | 0.85 | 0.0 | 0.0 | 0.85 | 0.0 | 0.0 |
| PyramidNet-110 | Epoch | 300 | 300 | 300 | 300 | 300 | 300 |
| | Batch size | 256 | 256 | 256 | 256 | 256 | 256 |
| | Data augmentation | Basic | Basic | Basic | Basic | Basic | Basic |
| | Peak learning rate | 0.10 | 0.10 | 0.10 | 0.10 | 0.10 | 0.10 |
| | Learning rate decay | Cosine | Cosine | Cosine | Cosine | Cosine | Cosine |
| | Weight decay | $5 \times 10^{-4}$ | $5 \times 10^{-4}$ | $5 \times 10^{-4}$ | $5 \times 10^{-4}$ | $5 \times 10^{-4}$ | $5 \times 10^{-4}$ |
| | $\rho$ | 0.2 | 0.2 | 0.2 | 0.2 | 0.2 | 0.2 |
| | $\alpha_{max}$ | 0.9 | 0.0 | 0.0 | 0.9 | 0.0 | 0.0 |

Table 7: Hyperparameters for training from scratch on ImageNet.

| | ResNet-50 | | | ResNet-110 | | |
|---|---|---|---|---|---|---|
| | $\epsilon$-MS | SAM | ESAM | $\epsilon$-MS | SAM | ESAM |
| Epoch | 90 | 90 | 90 | 90 | 90 | 90 |
| Batch size | 512 | 512 | 512 | 512 | 512 | 512 |
| Peak learning rate | 0.2 | 0.2 | 0.2 | 0.2 | 0.2 | 0.2 |
| Learning rate decay | Cosine | Cosine | Cosine | Cosine | Cosine | Cosine |
| Weight decay | $1 \times 10^{-4}$ | $1 \times 10^{-4}$ | $1 \times 10^{-4}$ | $1 \times 10^{-4}$ | $1 \times 10^{-4}$ | $1 \times 10^{-4}$ |
| $\rho$ | 0.05 | 0.05 | 0.05 | 0.05 | 0.05 | 0.05 |
| Input resolution | $224 \times 224$ | $224 \times 224$ | $224 \times 224$ | $224 \times 224$ | $224 \times 224$ | $224 \times 224$ |
| $\alpha_{max}$ ' | 0.8 | 0.0 | 0.0 | 0.7 | 0.0 | 0.0 |

Table 8: Hyperparameters for long-tail training for GLMC+SAM and GLMC+$\epsilon$-MS.

| | CIFAR-10-LT | | CIFAR-100-LT | |
|---|---|---|---|---|
| | 0.02 | 0.01 | 0.02 | 0.01 |
| Epoch | 200 | 200 | 200 | 200 |
| Batch size | 64 | 64 | 64 | 64 |
| Peak learning rate | 0.01 | 0.01 | 0.01 | 0.01 |
| Learning rate decay | Cosine | Cosine | Cosine | Cosine |
| Weight decay | $5 \times 10^{-3}$ | $5 \times 10^{-3}$ | $5 \times 10^{-3}$ | $5 \times 10^{-3}$ |
| $\rho$ | 0.05 | 0.05 | 0.05 | 0.05 |
| $\alpha_{max}$ | 0.2 | 0.2 | 0.1 | 0.2 |

Table 9: Hyperparameters for Single domain adaptation

| | PACS | | Office-Home | |
|---|---|---|---|---|
| | SAM | $\epsilon$-MS | SAM | $\epsilon$-MS |
| Epoch | 50 | 50 | 50 | 50 |
| Batch size | 64 | 64 | 64 | 64 |
| Peak learning rate | 0.0001 | 0.0001 | 0.0001 | 0.0001 |
| Learning rate decay | Cosine | Cosine | Cosine | Cosine |
| Weight decay | 0 | 0 | 0 | 0 |
| $\rho$ | 0.05 | 0.05 | 0.05 | 0.05 |
| $\alpha_{max}$ | 0 | 0.65 | 0 | 0.5 |