# OpenReview forum: "Rethinking Flat Minima: Seeking $\epsilon$-Maxima Towards Better Generalization"
_ICLR.cc/2026/Conference — Submitted to ICLR 2026_

### Official Review · Reviewer_E69K · 2025-10-15

**Soundness:** 2
**Presentation:** 3
**Contribution:** 2
**Rating:** 2
**Confidence:** 4

**Summary:**

Following the observation that flat minima seem to empirically generalize better than sharp minima, the paper proposes taking this a step further by finding local maxima- points that are higher on the training loss landscape than their surroundings. The paper proposes an algorithm to find such maxima, which boils down to adding a term to SAM that goes in the direction of the original, unperturbed gradient. Experiments show the proposed method outperforms baselines.

**Strengths:**

The paper poses a very interesting premise that provides a fresh perspective on shapness-aware algorithms. The derivation of the proposed algorithm is sound, and on the plus side the algorithm itself is simple and efficient to implement.

The paper certainly has the potential to be of interest to those in the sharpness community.

**Weaknesses:**

Unfortunately, the paper has weaknesses on both the conceptual side and empirical side. Conceptually, while the motivation behind a maximum finding algorithm is really intriguing, the proposed algorithm boils down to a very simple modification of SAM that adds an additional unperturbed gradient term. By itself, I'm not even sure if this is fully novel- if alpha were fixed, this algorithm could likely be found in prior literature. Nevertheless, the authors propose an adaptive alpha which does make this contribution more original. However, the connection between finding local maxima and the proposed algorithm is tenuous. It's not clear at all whether the algorithm's better performance is due to it finding local maxima or due to some other factor (such as being better at finding flatter regions of a loss landscape). Indeed, Figure 3d seems to show that the proposed algorithm just finds a flatter region than SAM. Rather than simply assessing the performance of the epsilon-MS algorithm, it would be much more interesting to directly study whether well-generalizing parameter values actually obey equation 6 (and for which values of q, \rho and \epsilon).

Given that the paper has no strong theory component (beyond equation 7), the empirical results are critical. Unfortunately, the evaluation could be stronger. The evaluations on CIFAR are good: the authors compare with a number of baselines over multiple trials. However, on Imagenet, very few baselines are considered and only a single trial is reported. This makes it hard to assess whether the proposed method is state-of-the-art as well as the statistical significance of the results.

Minor points:
- Figure 2, 3ab are too small
- Figure 2 is quike spiky and hard to interpret even when zoomed in; I recommend smoothing over epochs (e.g. using a sliding window)
- Figures should include error bars over multiple trials where applicable
- Figure 4 has no caption

**Questions:**

- Please add at least two additional baselines on Imagenet
- See minor points above

---

### Official Review · Reviewer_u4Lw · 2025-10-30

**Soundness:** 3
**Presentation:** 3
**Contribution:** 3
**Rating:** 8
**Confidence:** 3

**Summary:**

This paper challenges the conventional belief that flat minima necessarily yield superior generalization in deep neural networks. The authors argue that, in highly over-parameterized models, flat minima may not always correspond to optimal generalization. They introduce the concept of $\epsilon$-Maxima, a geometric structure where the central point’s loss is slightly higher than its surrounding neighborhood, forming a “peak-with-moat” configuration that improves population-level robustness. Building on this, the proposed $\epsilon$-Maxima Seeking algorithm constrains the central loss to decrease more slowly than the worst-case neighborhood loss via an adaptive coefficient $\alpha$, maintaining a positive margin and reducing the PAC-Bayesian generalization bound. Extensive experiments validate the effectiveness of the proposed method.

**Strengths:**

- The paper is generally well-written and easy to follow.

- This paper rethinks SAM’s pursuit of flatter minima and proposes a novel approach that instead seeks $\epsilon$-Maxima, which appears to contradict the traditional view of finding flat minima as in SAM. The idea is novel and interesting, and the explanation successfully clarifies why the method is sound.
- The paper is theoretically grounded. It provides a PAC-Bayesian generalization bound indexed by the punctured neighborhood, establishing a formal connection between the optimization objective and generalization performance, which strengthens the theoretical solidity of the proposed method.
- The experiments cover a broad range of tasks, from standard classification to long-tailed recognition and single-domain generalization, demonstrating the robustness and effectiveness of the proposed method across diverse settings.

**Weaknesses:**

- For the long-tailed learning experiments, the evaluations are mainly conducted on small-scale datasets such as CIFAR-10-LT and CIFAR-100-LT. It would be beneficial to include experiments on larger-scale long-tailed datasets, such as ImageNet-LT or Places-LT, to further validate the effectiveness and scalability of the proposed method.

- More representative SAM variant methods and long-tailed SAM-based methods, such as Fisher SAM [1], GAM [2], ImbSAM [3], CC-SAM [4], and Focal-SAM [5] should be included in the related work for a more comprehensive review.
- There are also some typos, such as in line 778 in Appendix A.5, where "we use apply the adaptive $\alpha$" should be corrected to "we use the adaptive $\alpha$"

-----

[1] Fisher SAM: Information Geometry and Sharpness Aware Minimisation, ICML 2022

[2] Gradient Norm Aware Minimization Seeks First-Order Flatness and Improves Generalization, CVPR 2023

[3] ImbSAM: A Closer Look at Sharpness-Aware Minimization in Class-Imbalanced Recognition, ICCV 2023

[4] Class-Conditional Sharpness-Aware Minimization for Deep Long-Tailed Recognition, CVPR 2023

[5] Focal-SAM: Focal Sharpness-Aware Minimization for Long-Tailed Classification, ICML 2025

**Questions:**

Please see above.

---

### Official Review · Reviewer_m6mx · 2025-10-31

**Soundness:** 3
**Presentation:** 2
**Contribution:** 2
**Rating:** 4
**Confidence:** 4

**Summary:**

The paper proposes to seek \epsilon-maximum points instead of flat minimum in the optimization for deep neural networks, i.e., find a punctured flat valley where result should be in the middle \epsilon-higher region. The approach is motivated through PAC-Bayes derivation based on SAM derivation and compared to multiple SAM analogues on different tasks, including long-tailed ones and out-of-domain generalization. The practical implementation (as in the case of SAM as well) goes away from the derived quantities, approximating the gradient with mixture of the local maximum and current point, where mixture is adaptive during training according to the difference of the strength of the gradients.

**Strengths:**

The paper analyses alternative approach to achieving better generalization than flat minimum, performing interesting theoretical and empirical analysis, which checks the performance on non-standard challenging tasks.

**Weaknesses:**

I find the motivation behind the proposed geometrical structure of the desirable minimum somewhat unclear. The manuscript states that “on the population it may move away and have loss lower than training”. However, if the solution lies within a flat valley where the loss remains equally low across neighboring points, and the population loss changes at a similar rate, the outcome would appear to be the same. Clarifying the intended distinction here would be helpful.

The paper also claims that overparameterized networks can drive the training loss arbitrarily low, even at a local maximum. This statement seems tautological, since by definition the lowest point of the loss function is its global minimum and cannot coincide with a maximum. In addition, the discussion in lines 131–136 appears somewhat outdated. First, regardless of overparameterization, standard training procedures in modern neural networks rarely lead to overfitting. Second, when considering sharpness from a Hessian-based perspective, sharp minima can generalize as well as flat ones, due to parameter reparameterizations and symmetries.

The manuscript also omits reference to established theoretical connections between generalization and flatness—such as those derived from PAC-Bayes theory [1] or more recent bounds [2]. While the abstract suggests these are not considered, the PAC-Bayes framework is later used in the analysis, which introduces some inconsistency.

Furthermore, the statement on line 152 appears incorrect: there cannot be a distributional discrepancy between the training set and the population under the assumption of i.i.d. samples. This assumption is fundamental to the standard machine learning setup used for both training and evaluation.

Finally, the paper claims to achieve state-of-the-art results on CIFAR-10 and CIFAR-100, yet the reported performance does not match that of the original SAM paper [1].

Minor comment:

- The expression “strong generalization scenario” is quite uncommon in the deep learning literature and might benefit from rephrasing.

[1] Foret, Pierre, et al. “Sharpness-Aware Minimization for Efficiently Improving Generalization.” ICLR (2021).

[2] Petzka, Henning, et al. “Relative Flatness and Generalization.” NeurIPS 34 (2021): 18420–18432.

**Questions:**

1 - The proposed approach appears to implicitly depart from the standard empirical risk minimization (ERM) framework. How do you position your method with respect to ERM?

2 - You claim that overparameterization leads to complex loss landscapes. However, overparameterization typically refers to the dimensionality of the loss surface rather than its complexity. Could you clarify what you mean by this statement?

3 - The paper mentions that the proposed approach is inspired by the ε-family in optimization and variational analysis, but this connection is not elaborated upon. Could you provide more detail on this relationship?

4 - If I understand correctly, your approach effectively mixes the gradient at the local maximum with the point itself to determine the update step. Is this interpretation accurate?

5 - In Figure 3(d), the resulting configuration does not appear to correspond to the desired “punctured valley” structure described in the text. How do you explain this discrepancy? The observed configuration seems more consistent with the asymmetric valleys characteristics discussed in [1], which are not cited in your work.

[1] He, Haowei, Gao Huang, and Yang Yuan. “Asymmetric Valleys: Beyond Sharp and Flat Local Minima.” NeurIPS 32 (2019).

---

### Official Review · Reviewer_wMT7 · 2025-10-31

**Soundness:** 3
**Presentation:** 3
**Contribution:** 3
**Rating:** 6
**Confidence:** 4

**Summary:**

The paper proposes an adaptation to the SAM objective that optimizes not for a low maximum loss in a neighborhood of the model, but rather for a maximum loss in the neighborhood that is smaller than the loss of the model. That is, it seeks for a point where the loss in the surrounding is smaller. The new optimization objective is effectively the SAM loss minus the loss at the current model. The paper proposes to weigh the loss at the current model adaptively through a parameter $\alpha$ that is set to guarantee that with each update step the margin between center loss on maximum loss in the neighborhood increases. The paper shows that this objective still reduces the center loss but slows it down, potentially preventing overfitting.

**Strengths:**

- Improving neural network optimization is always valuable and the proposed idea is novel and interesting.
- The idea of requiring a neighborhood with low loss to improve robustness is intriguing.
- The optimization problem is sound, the theoretical analysis is clean and connects to a PAC-Bayesian bound, which is rare in optimization-focused work.
- The paper is very well written and easy to follow.

**Weaknesses:**

- The empirical improvements over SAM are very small and in many cases not statistically significant (Tab 1: CIFAR10 ResNet-18, and WRN-28-10, Tab 3: CIFAR-10-LT 100 and 50, CIFAR-100-LT 100). I am aware, though, that beating SOTA nevertheless is a notable achievement.
- Missing related work: Due to the _reparameterization-curse_ [3], several reparameterization-invariant flatness measures have been proposed, e.g., the Fisher-Rao-Norm [5] and Relative Flatness [6]. The latter has been theoretically linked to generalization [7, 4] and adversarial robustness [8], and it was used in optimization [1].
- More recent work has shown that SAM does not lead to flatter solutions [Andriushenko, Wen], but instead seems to rather improve the conditioning of the optimization problem. Therefore, it is unclear how the SAM objective in $\epsilon$-MS improves robustness. That is, while the $\epsilon$-MS objective guarantees some robustness wrt. the center model by definition, it is unclear how this compares to a flat solution according to standard flatness measures, such as Fisher-Rao-Norm or Relative Flatness.
- It is not clear whether $\epsilon$-MS actually optimizes some form of flat geometry (or moat-like geometry) or merely re-weights gradient directions to slow center descent.

**Questions:**

- How important is the dynamic $\alpha$? That is, how well would $\epsilon$-MS work with a fixed $\alpha$? An ablation would be interesting.
- Similarly, how does $\alpha$ behave during training? That is, does the training always have an $\alpha>0$, or does it boil down to SAM at times?
- What is the value range for the heatmap in Fig. 3 c and d? And isn't the heatmap of $\epsilon$-MS counter-intuitive? I would have expected a larger loss value at the center, similar to the moat analogy. Does that imply that, similar to SAM, the intuition of what $\epsilon$-MS should do does not fit what it actually does?
- The introduction of SAM (Foret, et a., 2020) is not really recent (line 40)., right?
- Could the authors quantify curvature (e.g., via top Hessian eigenvalues) or compare to reparameterization-invariant flatness metrics to clarify what geometry $\epsilon$-MS actually yields?

**References:**

[1] Adilova, Linara, et al. "FAM: Relative Flatness Aware Minimization." Topological, Algebraic and Geometric Learning Workshops 2023. PMLR, 2023.

[2] Andriushchenko, Maksym, and Nicolas Flammarion. "Towards understanding sharpness-aware minimization." International conference on machine learning. PMLR, 2022.

[3] Dinh, Laurent, et al. "Sharp minima can generalize for deep nets." International Conference on Machine Learning. PMLR, 2017.

[4] Han, Ting, et al. "Flatness is Necessary, Neural Collapse is Not: Rethinking Generalization via Grokking." Advances in Neural Information Processing Systems, 2025.

[5] Liang, Tengyuan, et al. "Fisher-rao metric, geometry, and complexity of neural networks." The 22nd international conference on artificial intelligence and statistics. PMLR, 2019.

[6] Petzka, Henning, et al. "A reparameterization-invariant flatness measure for deep neural networks." Science meets Engineering of Deep Learning 2019. Neural Information Processing Systems (NIPS), 2019.

[7] Petzka, Henning, et al. "Relative flatness and generalization." Advances in neural information processing systems 34 (2021): 18420-18432.

[8] Walter, Nils Philipp, et al. "When Flatness Does (Not) Guarantee Adversarial Robustness." arXiv preprint arXiv:2510.14231 (2025).

[5] Wen, Kaiyue, Tengyu Ma, and Zhiyuan Li. "How Does Sharpness-Aware Minimization Minimizes Sharpness?." OPT 2022: Optimization for Machine Learning (NeurIPS 2022 Workshop).

---

### Meta-Review · Area_Chair_REJp · 2026-01-06

**Summary:**

This paper aims to investigate flat minima from a perspective different from Sharpness-Aware Minimization (SAM). While some reviewers appreciated this alternative viewpoint, others raised fundamental concerns, leading to highly divergent scores. The AC aligns with the negative assessments, particularly Reviewer E69K’s observation that “the proposed algorithm boils down to a very simple modification of SAM that adds an additional unperturbed gradient term.”

The experimental evaluation also exhibits critical weaknesses: experiments are limited to CIFAR datasets, and many strong and relevant baselines are missing.

Furthermore, the authors did not provide any rebuttal or clarification during the response period.

Overall, the final recommendation is REJECTION.

**Reviewer Concerns:**

This paper received extremely divergent scores. Two reviewers viewed the core idea positively, while the other two questioned whether the method is essentially a minor variant of SAM with an added gradient component. The AC supports this latter viewpoint and notes that a highly similar update strategy appears in T. Li et al., “Revisiting Random Weight Perturbation for Efficiently Improving Generalization,” TMLR 2024.

Regarding the empirical evaluation, three reviewers (with scores of 2, 4, and 6) raised substantial concerns, including the narrow experimental scope (restricted to CIFAR), the absence of strong baseline comparisons, and insufficient clarity regarding hyperparameter settings.

No author response was provided to address these concerns.

**Reviewer Scores:**

The initial score is 2/4/6/8. The authors did not provide any response during the rebuttal period.

---

### Decision · Program_Chairs · 2026-01-26

Reject